# Customer Analysis Using Machine Learning-Based Classification Algorithms for Effective Segmentation Using Recency, Frequency, Monetary, and Time

**DOI:** 10.3390/s23063180

**Published:** 2023-03-16

**Authors:** Asmat Ullah, Muhammad Ismail Mohmand, Hameed Hussain, Sumaira Johar, Inayat Khan, Shafiq Ahmad, Haitham A. Mahmoud, Shamsul Huda

**Affiliations:** 1Department of Computer Science, Brains Institute, Peshawar 25000, Pakistan; asmat227@gmail.com (A.U.); muhammadismail1745@gmail.com (M.I.M.);; 2Department of Computer Science, University of Buner, Buner 19290, Pakistan; dr.hameed@ubuner.edu.pk; 3Department of Computer Science, University of Engineering and Technology, Mardan 23200, Pakistan; inayatkhan@uetmardan.edu.pk; 4Industrial Engineering Department, College of Engineering, King Saud University, P.O. Box 800, Riyadh 11421, Saudi Arabia; 5School of Information Technology, Deakin University, Burwood, VIC 3128, Australia; shamsul.huda@deakin.edu.au

**Keywords:** recency, agglomerative, k-means, Gaussian, dbscan, silhouette, Calinsky–Harabasz, Davies–Bouldin, Dunn index, customer segmentation

## Abstract

Customer segmentation has been a hot topic for decades, and the competition among businesses makes it more challenging. The recently introduced Recency, Frequency, Monetary, and Time (RFMT) model used an agglomerative algorithm for segmentation and a dendrogram for clustering, which solved the problem. However, there is still room for a single algorithm to analyze the data’s characteristics. The proposed novel approach model RFMT analyzed Pakistan’s largest e-commerce dataset by introducing k-means, Gaussian, and Density-Based Spatial Clustering of Applications with Noise (DBSCAN) beside agglomerative algorithms for segmentation. The cluster is determined through different cluster factor analysis methods, i.e., elbow, dendrogram, silhouette, Calinsky–Harabasz, Davies–Bouldin, and Dunn index. They finally elected a stable and distinctive cluster using the state-of-the-art majority voting (mode version) technique, which resulted in three different clusters. Besides all the segmentation, i.e., product categories, year-wise, fiscal year-wise, and month-wise, the approach also includes the transaction status and seasons-wise segmentation. This segmentation will help the retailer improve customer relationships, implement good strategies, and improve targeted marketing.

## 1. Introduction

Business is always the result of demand from society and supply from business firms. Every industry’s focal point is its customers; industries always run around the needs of their customers. If a company is small or huge, it must compete with others. Many of the competitors are not succeeding. A business may fail for numerous reasons, but according to us, one of the most common causes of failure is “companies opting to avoid knowing their customers” Rahul, S. [1].

The cost of attracting new consumers is substantially higher than retaining existing ones. As a result, the most critical concern for businesses is how to sell more items to current clients. Using a platform’s purchase data to understand how users make decisions in the real world has become a fundamental challenge to tackle the efficient operation of businesses. Customer segmentation, in basic terms, is the process of separating consumers, marketing to them based on different criteria, and putting them together based on comparable qualities. As an outcome, each customer segment needs a unique marketing or strategic method.

The e-commerce market cape is growing with that. Online marketing grew in scope. There are more opportunities for companies and marketing persons to access customers digitally. Pakistan is also a vast market, and e-commerce is becoming popular in Pakistan. The country’s e-commerce market grew by 78.9% in volume and 33.3% in worth K, T.H. [2]. E-commerce income climbed drastically from PKR 2.3 billion to PKR 9.4 billion in the fourth quarter, increasing the yearly revenue to PKR 34.8 billion.

Understanding consumer attributes is a key to success in e-commerce and developing targeted marketing strategies for different types of customers (Jinfeng, Z.) [3]. For this purpose, we need customer-segmented data to target them for marketing. In this study, the researcher used Pakistan’s largest e-commerce dataset (Zeeshan-ul-Hassan, U.) [4] to assist new and existing businesses in Pakistan.

Recency: The most recent transaction date is deducted from the specified date, and the result is expressed in months. Frequency: The number of transactions per consumer. Monetary: The monetary worth of each transaction is added together for each consumer. Time: The number of days between successive transactions is summed, then converted to months.

In this article, the authors analyzed the RFMT dimensions of the customers. Clustering analysis factors are considered from cluster 0 to cluster 10, using the elbow, dendrogram, silhouette, Calinsky–Harabasz, Davies–Bouldin, and Dunn index. From cluster factors analysis, the stable cluster is elected through majority voting, which results in 03 for Gaussian, hierarchical, and k-means, and 02 for DBSCAN. Segmenting data using various machine learning algorithms such as Gaussian, hierarchical, k-means, and DBSCAN were used. The dataset was additionally segmented on these algorithms based on the payment method, transaction status, product type, month of purchase, financial year, and purchase seasons.

The seller can increase their profit from strategies adopted for targeting customers according to their needs and habits and by providing different packages identified in the customer segmentation process.

### 1.1. Contributions

The largest Pakistani e-commerce dataset was used and segmented based on payment methods, transaction status, product type, purchase month, financial year, and session purchases. The RFMT model was applied to the dataset and different techniques were used to determine the number of clusters.A cluster analysis was performed using a variety of parameters.

In this research article, we have used cluster validation criteria to verify the cluster’s validity, majority voting to select the cluster, and using different algorithms for segmentation on the RFMT model.

### 1.2. Paper Organization

The residual parts of this paper are organized as follows: Section 1 discusses the introduction about the model and follows the contribution of the research work. In Section 2, the relevant studies focus on customer segmentation, algorithms, RFM models, inter-purchase time T, and majority voting. In Section 3, the methodology customer segmentation framework is described. In Section 4, the results and discussion are shown. Section 5 is the conclusion of this research study.

## 2. Related Works

### 2.1. Customer Segmentation

Consumer segmentation is splitting all consumers into distinct groups based on features such as tariff plan, network voice, smartphone apps, invoicing, network information, shops, cell center, webpage, and roaming. It can help the trades focus marketing struggles and resources on valuable, loyal consumers to meet the trades Ioannis, M. [5]. In Sukru, O. [6] and Himanshu, S. [7], the authors performed customer segmentation using machine learning techniques; their main point was customer happiness and brand choice, respectively. The aims were achieved using k-means, hierarchical clustering, density-based clustering, and affinity propagation Aman, B. [8]. A comparative dimensionality reduction study Maha, A. [9] was conducted. The authors performed customer segmentation to reduce 220 characteristics for 20 features for 0.1 million customers by using a k-means clustering algorithm with principal component analysis. In Dong [10], the authors studied brand purchase prediction by exploring machine learning techniques. The three primary duties in this review research were predicting customer sessions, purchasing choice, and customer desire. A data-driven solution that only requires part knowledge of the target regions has been created to address the models. This technique presents a data-collecting method of Points of Interest (POIs), a clustering-based method that can be used to pick alternative refueling stations Ge, X. [11].

Businesses can gain a better understanding of their customer base and identify valuable, loyal customers. This can lead to more effective marketing campaigns and increased customer satisfaction.

The whole dataset produced 175 features in this study to identify the stable cluster. On these features, this study performed clustering and segmentation.

### 2.2. Algorithms

Gaussian is used to minimize various drawbacks, including noise and accuracy problems. In Ting, Z. [12], the author used Gaussian with the combination of fuzzy-C mean clustering for segmentation purposes; therefore, in this study, the internal factors for cluster analysis are performed through k-means, agglomerative hierarchy, DBSCAN, and SOM and compared on four datasets. As a result, the best-performing cluster algorithm is identified for each dataset Abla, C.B. [13]. The k-means algorithm performs well when the data are as significant as retrieved from the disk and stored in the primary memory. The k-means quickly result when the data are big M, S. [14]. Xin, S.Y. [15] When all the clusters are formed, the maximum distance is permitted between the clusters. A horizontal line is plotted, which passes through the dendrogram plot; the number of cuts represents the number of clusters.

In this work, multiple algorithms k-means, agglomerative, Gaussian, and DBSCAN, were used to cluster data; these algorithms took the stable cluster value. Each algorithm used its own characterized approach to perform segmentation.

### 2.3. RFM (Recency, Frequency, and Monetary) Analysis

In Rajan, V. [16], specific audiences were targeted; in Saurabh, P. [17], startup businesses assessed their customers; Rahul, S. [1] looked at buying data from September to December 2018 to compute indicators that enhanced RFM; and Jun, W. [18] identified customers to design promotional activities; all these used k-means and RFM model.

In Onur, D. [19], the number of clusters, or K value, was calculated using the silhouette approach. In P, A. [20], the segmentation was performed using the RFM model and K-means to quantify electronic industry data. The entropy factor for cluster factor analysis is used to find and choose the best cluster; the performance of k-means is the most extensively used partition clustering technique, Ching, H.C. [21].

The RFMT-purchased data collections are mapped into distinct groups called RFMT scoring. In this paper, there are two quintiles of scoring discussed. They are customer quintile and behavior quintile scoring. The frequency and monetary values of the records are ordered in ascending order and then divided into five quintiles or groups.

The RFM (Recency, Frequency, and Monetary) analysis is a widely used approach in customer segmentation; it does not consider an essential factor of time, i.e., T. Thus, our research could investigate the inclusion of time (T) in the RFMT model to better understand customer loyalty and customer behavior. So, taking this into account brings long-term relationships with customers.

### 2.4. Inter-Purchase Time

The time difference between two successive transactions for the same customer in the dataset is the inter-purchase time, T. Since the 1960s, this method has been used in business for behavior analysis Donald, G.M. [22]. The consistency and tendency of the customers towards shopping behavior were studied and used T. Similarly, the T checks customer reliability and trustworthiness in their purchasing behaviors Demetrios, V.; Lars, M.W. [23,24]. Introducing the multi-category T model that predicts customer buying behavior, Ruey, S.G. [25] developed the multi-category T model to increase product recommendations effectively Junpeng, G. [26].

T was also introduced for customer segmentation. The RFMT model is the complete model for analyzing consumers’ purchase groups over an extensive duration, using an algorithm with results that may narrow the segmentation approach. We used the RFMT model and applied a novel approach for segmentation Jinfeng, Z. [3].

### 2.5. Internal Cluster Validation

The intra-cluster distances were minimized while increasing inter-cluster distances: silhouette, the Dunn index, the Calinski–Harabasz index, and the DB index can be used to validate the clusters.

Ref. [1] used the silhouette and elbow methods, ref. [3] used Calinski–Harabasz and Davies–Bouldin, while Xin, S.Y. [13] used Dunn.

No validation or validation on only one criterion may be biased or may produce biased results. The literature review suggests that different cluster validation factors have been used to validate the clusters (silhouette, Calinski–Harabasz, Dunn index, Davies–Bouldin, and Dendrogram), but there is no agreement on which factor is the most effective.

We used silhouette, the Dunn index, the Calinsky–Harabasz index, and the Davies–Bouldin index of internal cluster validation factors in this research work. Using a variety of validation factors instead of one factor will lead to accurate clustering of the data.

### 2.6. Majority Voting

Because of the different characteristics of the algorithms, it might be challenging to choose the right cluster. The cluster for the model is selected by a majority vote Donald, G.M. [27]. The challenge is choosing the best segmentation approach due to the different characteristics of the algorithms. Thus, our research could investigate the use of majority voting, an ensemble method that combines multiple clustering algorithms, to improve the accuracy and stability of customer segmentation.

We used the majority voting-based novel approach for an RFMT-based clustering model.

## 3. Methodology

The proposed framework, Figure 1, defines the architecture of the customer segmentation system. An e-commerce dataset is loaded into the system, and data preprocessing is performed. The first step removes null, missing, and invalid literals. Then, the string is converted to numbers and dates as required. In the loaded dataset, there were 584,524 records in 21 attributes. After preprocessing the data, this research refined the dataset with 582,241 catalogs in 21 attributes. The quintile score is predefined for recency, frequency, monetary, and time. The CustomerID then groups the data, so the total records after grouping are 115,081. The quintile scores are assigned to the grouped records. Each RFMT variable has a score for the grouped records. The RFMT is processed further and extracted, so the standard features are 175 × 4. Applying the elbow and dendrogram methods to the standard features gives us the cluster value, the cluster analysis factors silhouette, Calinski–Harabasz, Davies–Bouldin, and Dunn index from cluster 2 to 10 for different algorithms, i.e., k-means, agglomerative, and Gaussian are applied on standard features.

In addition, the cluster analysis factors silhouette, Calinski–Harabasz, Davies–Bouldin, and the Dunn index for ϵ values (1.93, 2.23, and 3) for DBSCAN is applied to standard features. It gives a stable weight for clusters, i.e., 2. Through majority voting and the statistical mode function, the cluster value is chosen. k-means, agglomerative, Gaussian, and DBSCAN are applied to the standard features data on the specified number (DBSCAN does need the cluster value) selected by majority voting. The RFMT with different algorithm cluster values for k-means, agglomerative, Gaussian, and DBSCAN is then applied to the grouped records and the primary dataset.

### 3.1. Dataset

This study used the largest Pakistani w-commerce dataset by Zeeshan-ul-Hassan, U. [4], containing data from 1 July 2016, to 28 August 2018. There are 21 fields in the dataset and half a million transaction records. The fields we tackle are ‘Status’, ‘created_at’, ‘price’, ‘MV’, ‘grand_total’, ‘category_name’, ‘payment_method’, ‘year’, ‘month’, ‘FY’, and ‘Customer ID’. The transaction status value is either completed, incomplete, canceled, or refunded, etc., as we segmented the data based on the status field. Therefore, the field is selected. ‘Created_at’ (the sale date) provides information about the transactions that have occurred to date, and the time is calculated from this field. ‘Price’ gives information about the product price. ‘MV’ is monetary or the actual price paid for the product. ‘Grand_total’ is the total paid value of a transaction. ‘Category_name’ (category of the product) gives information about the product category to which it belongs. The ‘payment_method’ field shows the method of payment for the product. The ‘year’ field gives information about the year on which the product transaction occurred. The ‘month’ field provides information about the month in which the product transaction occurred. ‘M-Y’ (month and year) is the month and year of the transaction. ‘FY’ (financial year) shows the transaction’s financial year. ‘Customer ID’ is the unique ID of the customer.

The tool used is Python 3.8.5 Jupiter Notebook. The dataset is chosen to analyze and benefit the local market businesses.

### 3.2. Data Preprocessing

This section performs data preprocessing before feeding it to the proposed machine learning model. Null, negative, missing, and invalid literals are removed during data cleaning. Through the RFMT model, customer segmentation is performed; therefore, it must translate data from the obtained dataset to the RFMT data pattern. Initially, the Customer ID is a one-of-a-kind identifier that serves as the primary key. The column names are ‘created at’ for recency, ’increment id’ for frequency, ‘MV’ for monetary and ’WorkingDate’ for time. The RFMT values of the associated customer from the dataset are computed and renamed for a specific ID. The monetary (M) value was calculated using all the expenses from the particular customer. The frequency (F) value was calculated using the number of purchases made by the customer. The recency (R) value was calculated using the time gap between the customer’s recent purchase and the drawn date, 1 March 2020. The months were the unit of time in this study, while used for recency and time. Enter purchase duration (T), the fourth variable, measures the average time between successive purchasing transactions. If a customer’s initial and final purchase dates are *t*_1_ and *t_n_*, the customer’s rounded purchasing cycle (T) may be estimated by the months between *t*_1_ and *t_n_*, and so the T (in the months) can be computed as follows: to compute T, use the formula:(1)$$T\;=t_{n}-t_{1},$$

The dataset had 584,524 shopping records from 115,081 distinct consumers. After data preprocessing, Table 1 evaluates the transaction records for three customers (CustomerID: 02, 03, and 04).

### 3.3. RFMT Criteria for Scoring

The dataset values, the numbers at different centiles, and the number of transactions for recency, frequency, monetary, and time are given in Table 2.

UB is the upper boundary value for a specific centile. This is a system-generated value for RFMT variables. Following a specific translating rule, the RFMT results are translated into a 5-quintile scale. Table 3 shows the results. Recency (18.12, 44), frequency (1, 2524), monetary (1, 36,202,688), and inter-purchase time (0, 25), respectively, are on various units/unit-less and have highly distinct data collections. Before the clustering analysis, these variables should be uniformly scaled or discretized. The study followed the John, R.M. [28] rating guidelines for creating monetary and frequency quintiles. The last transaction in the dataset is 28 August 2018, and the withdrawal date was chosen as 1 March 2020. The lower value of recency and time attributes will produce a higher score, i.e., if the transaction lies in 20 centiles, it will produce 5 scores, 40 centiles will make 4, 60 centiles produce 3, 80 centiles have 4, and over 80 centiles will give the value of 5 for both R and T. For the F and M quintiles: score 1 = 20 centiles, score 2 = 40 centiles, score 3 = 60 centiles, score 4 = 80 centiles, and score 5 = >80 centiles for each F and M. Table 3 presents the scoring procedures for RFMT discretization on a quintile scale. Using the data from Table 1, Table 2 and Table 3 shows the discretized scores for the three customers and depicts the RFMT distributions across the discretized scale extracted from the values in Table 1 and Table 2.

### 3.4. Data Mining

#### 3.4.1. Elbow Method

The elbow approach calculates the optimum number of clusters based on recency, frequency, monetary, and time. The sum of squared errors (SSE) is shown against a reasonable number of cluster values. The chosen value at the graph’s maximum curve is called the K value.

#### 3.4.2. Silhouette Score

The silhouette value varies from −1 to +1, with a high value representing a well-matched item and a low one showing the opposite. The silhouette index helps determine the correct cluster design; for example, if many points are low or negative, the clustering arrangement may have many or few clusters Figure 2 shows the silhouette coefficient for different algorithms used in this study. The formula for the silhouette score is:(2)$$S_{\left(i=2\;to\;n\right)}=(S_{i}-{S^{\prime }}_{i})/Max(S,{S^{\prime \prime }}_{i}),$$
where:

$S_{i}$ = Average distance of items between *i*th group/cluster.

${S^{\prime }}_{i}$ = Average distance between *i*th cluster with different groups/clusters.

$Max(S_{i},{S^{\prime }}_{i})$ = Average distance between $S_{i\;}$ with ${S^{\prime }}_{i}$.

#### 3.4.3. Calinski–Harabasz and Davies–Bouldin

Calinski–Harabasz: A higher CH index indicates that the clusters are dense and well-spaced. Figure 2 shows the Calinski–Harabasz value for the different algorithms used in this research; nevertheless, if the line is uniform (horizontal, rising, or descending), there is no reason to choose one solution over another. The Davies–Bouldin index value decreases in direct proportion to the quality of the grouping. Figure 2 indicates the Davies–Bouldin value for the different algorithms related to this study. It does, however, have a downside. The low cost of this technique does not imply that it will provide the most effective information retrieval.

#### 3.4.4. Dunn Index

The greater the value of the Dunn index, the more significant the clustering is deemed to be. The ideal number of clusters, denoted by the letter k, is the number of groups that provide the highest Dunn index; in Figure 3, the author presented the Dunn index value for different algorithms.

#### 3.4.5. Dendrogram for Hierarchical Clustering

The graphical depiction of the hierarchical tree is called a dendrogram. The output in a dendrogram is a tree-based representation of the items presented in Figure 3. In this work, a dendrogram value for the optimal cluster is selected; that is, 03.

### 3.5. Machine Learning Models

#### 3.5.1. K-Means Clustering

The unsupervised ML approach, k-means clustering, is used to find groupings of data items in a dataset. Through k-means, we categorize k groups of similarity using Euclidean distance. The k-means algorithm is used with several clusters obtained in the elbow methods. The resulting output is shown in Table 4. When choosing the value of k, it is vital to remember that the “elbow” approach does not perform well with data that is not tightly grouped. A smooth curve is formed in this scenario, and the outstanding value of k will be ambiguous Martin, E. [29].

#### 3.5.2. Hierarchical Clustering

In this case, the K value is 3, as shown in the dendrogram diagram in Figure 3. The study uses agglomerative hierarchical clustering based on the bottom-up method. Using this technique, the study designated each data point belonging to a distinct cluster, quickly connected by merging the two most comparable groups. The cluster is decided to find suitable marketing tactics based on a high Calinski–Harabasz score and a relatively low Davies–Bouldin score if the group’s factors are variable.

#### 3.5.3. Gaussian

Gaussian mixture models (GMMs) are models based on the assumption that a set of Gaussian distributions exists, each representing a cluster of observations. As a result, it is related to the identical distributions clustered together in a Gaussian mixture model than in a normal distribution. Clusters of various sizes and correlation patterns can be accommodated using GMM clustering. Before fitting the model with GMM clustering, you must define the number of clusters. The number of groups in the GMM determines the number of components.

#### 3.5.4. Density-Based Spatial Clustering of Applications with Noise (DBSCAN)

The density of the data points in a zone determines cluster classifications. Where low-density areas separate large concentrations of data points, clusters are allocated. Unlike the other clustering methods, this method does not need the user to provide the number of clusters. Instead, there is a configurable threshold based on a distance-based parameter. This value controls how near points must be for them to be deemed cluster members. There are no centroids in Density-Based Spatial Clustering of Applications with Noise (DBSCAN); clusters connect neighboring points. However, it requires the input of two parameters that impact whether or not to connect two adjacent points into a single cluster.

Epsilon (ϵ) and min_Points are two different types of points. DBSCAN generates a circle with an epsilon radius around each data point and categorizes them as Core points, Border points, or Noise based on the circle’s radius. A data point is considered a Core point if the circle around it contains at least the specified number of points (min_Points). If the dataset has several dimensions, the value of min_Points should be larger than the number of dimensions, i.e., Martin, E. [30].
(3)min_Points ≥ Dimensions+1,

## 4. Results and Discussion

When a company has a thorough grasp of each cluster, it may build more tailored marketing approaches for particular consumer segments, resulting in more excellent customer retention. In all types of businesses, understanding the characteristics of each cluster group with the help of clustering can support the business professional and marketing persons to adopt more enhanced marketing strategies to target each customer segment for better operations. The different RFMT features in each cluster for other algorithms are analyzed in this section.

### 4.1. Cluster Value

The cluster value should be chosen using the dendrogram (Figure 3) and elbow method (Figure 2). Through elbow K = 4 and dendrogram = 3, the performance of the cluster models is validated and explained below.

### 4.2. Internal Cluster Validation

Cluster models are intended to minimize intra-cluster distances (distances between items within the same cluster) while increasing inter-cluster distances (distances between objects in other clusters) between objects inside other clusters. The following metrics are used to assess cluster model performance.

#### 4.2.1. Silhouette Width

This scale represents the distance between a cluster’s point and the other clusters’ points. It is between 0 and 1, with 1 representing well-clustered data. The following table, Table 4, shows the silhouette widths for the three cluster models.

#### 4.2.2. Dunn Index

The Dunn index is the ratio of the minimum inter-cluster length to the enormous intra-cluster length in a given cluster. A higher value of the Dunn index is ideal.

#### 4.2.3. The Calinski-Harabasz Index

The Calinski-Harabasz Index is a cluster validation index utilized internally by the cluster validation algorithm. Known alternatively as the Variance Ratio Criterion, the CH Index (also known as the Cohesion Index) is a statistic that compares how similar an item is to its cluster (cohesion) with other objects in other clusters (separation). The lengths between a group’s data points and the cluster’s centroid determine the group’s cohesiveness. On the other hand, the distance between cluster centroids and the global centroid is used to measure separation. The higher the CH index, the denser and more well-separated the clusters are.

#### 4.2.4. The Davies–Bouldin (DB) Index

The DB index is an internal evaluation method. The more acceptable the clustering, the lower the value of the DB index value becomes. It does, however, have a downside. The excellent value of this strategy does not imply that it will provide the most suitable information retrieval.

#### 4.2.5. Validation Matrics

Customer segmentation validation metrics Table 5 are used to evaluate the effectiveness and accuracy of the segmentation process for 10 clusters. Here we used Homogeneity, Silhouette score, Cohesion and Separation. As different factors for different algorithms result different clusters, therefore, we applied the majority voting to choose the appropriate cluster. That results in C3.

### 4.3. Majority Voting

The method of ensemble decision is known as majority voting. There are three varieties of it. When all classifiers agree, this is called unanimous voting. Simple voting is predicted by more than half of the classifiers. The candidate that receives the most votes is k-means = 3, hierarchical = 3, Gaussian = 3, and DBSCAN = 2 for ϵ = 2.23. The factors predicted by the clusters are (3, 7, 3, 8, 3, 3, 3, 9, 3, 5, 3, 8, 2, 2, 2, 2), take the frequency of each cluster value is
(4)$$f_{cluster}=(\mathrm{Number}\;\mathrm{of}\;\mathrm{Occurrences}\;\mathrm{of}\;\mathrm{the}\;\mathrm{cluster})\;$$

As Table 6 shows, *f*_3_ = 7 times, *f*_2_ = 4 times, *f*_5_ = 1 time, *f*_7_ = 1 time, *f*_8_ = 2 times, and *f*_9_ = 1 time.

The many factors for cluster analysis are listed below. Because of the component differences, choosing the right cluster might be challenging. As a result, the cluster for the model is selected by a majority vote. The cluster number for each algorithm is determined here.
(5)$${\mathrm{Model}}_{\mathrm{algo}}=\;\mathrm{Mode}\left({\mathrm{Silhouette}}_{\mathrm{algo}},{\mathrm{DI}}_{\mathrm{algo}},{\mathrm{CH}}_{\mathrm{algo}},{\mathrm{DB}}_{\mathrm{algo}}\right)$$
where algo is the algorithm, DI = Dunn index, CH = Calinski–Harabasz, DB = Davies–Bouldin.

They choose the optimum cluster, i.e., *f*_3_ = 7 times, because of the majority voting. As indicated in Table 4, DBSCAN has a marginally higher silhouette width than k-means, hierarchical, and Gaussian models. It should be noted that k-means, hierarchical, and Gaussian were built with three clusters, whereas DBSCAN was constructed only with two clusters. The two groups are not so deep to obtain the desired results while considering the dataset evaluation. Therefore, three clusters are elected.

The three clusters have 115,081 consumers and PKRS.4195251105 purchases over 26 months. Agglomerative, k-means, Gaussian, and DBSCAN clusters (C0) have a proportion of customers (37%, 18%, 18%, and 81%), respectively; cluster C1 has a proportion of customers of 18%, 32%, 43%, and 18%. Cluster C2 except DBSCAN has the proportion of customers of 43%, 49%, and 37%. Agglomerative and DBSCAN have a 54% share of the 4,195,251,105 total value, whereas k-means and Gaussian also have a 54% share. The average frequency for agglomerative and DBSCAN in C1 is 16 each, while k-means and Gaussian in C0 have 16 each of the 194,080 of the total frequency, the agglomerative C2, DBSCAN C0, and Gaussian C1 have the lower frequency value, i.e., 1. The agglomerative C1 has an average high recency value of 32, while k-means C0, Gaussian C0, and DBSCAN C1 have a lower recency value of 27.

The agglomerative average time is distributed in each cluster, while the other algorithms have 0 values in some clusters. Recency–frequency–monetary (RMF), inter-purchase time–frequency–monetary (TFM), and inter-purchase time–recency–monetary (TRM) graphs are used to create a three-dimensional (3D) representation of the data. Each diagram in Figure 4 depicts the relationship between three of the four variables (RFMT) in a specific cluster for the agglomerative, DBSCAN, Gaussian, and k-means models, as well as the relationship between three of the four variables (RFMT) in a given cluster.

### 4.4. Cluster C0, C1, and C2 of Different Algorithms

Gaussian and k-means show the same values as in Table 7 for the cluster (C0) and higher monetary value. The recency value of the DBSCAN (C0) is higher. The monetary value of the agglomerative (C0) is lower among all. As shown in Table 7, the time value for Gaussian and k-means is higher; DBSCAN contains the higher time value. The number of records for DBSCAN in T is 93,445. Time 0 means a higher quintile value, i.e., 5. K-means and Gaussian have the same values, while agglomerative and DBSCAN have time values that are 0, representing a higher value; that is, quintile value 5. The time (T) is higher in Table 7. The time gap is minor among all customers’ transactions; the summary of C0, C1, and C2 is shown in Table 7.

In cluster C2, the recency values for k-means are from mid to high. The customer’s records occurred in the mid towards high (Figure 4). For the k-means, the frequency lies at low and middle. The agglomerative has a low frequency value while the Gaussian frequency occurs from mid to high. The time value for k-means and Gaussian has the same value, with 0 having a high quintile value.

### 4.5. Summary of the Agglomerative, Gaussian, K-Means, and DBSCAN

The agglomerative in Figure 5B shows the three clusters graph in 5-quintile. The DBSCAN has two clusters categorizing the values in low and high (Figure 5D, recency). The Gaussian has three clusters with recency (Figure 5C, variations). The k-means recency varies from cluster to cluster and quintile to quintile (Figure 5A).

The summary of the agglomerative, Gaussian, k-means, and DBSCAN are shown in Table 7. The tables contain the number of customers (#Customer), monetary, frequency, recency, and time values for different clusters.

### 4.6. Status Analysis by Clusters

Table 8 is the tabular description of the data which shows the transaction status across different clusters and algorithms that most transactions are completed across each cluster. The \N shows the null transactions.

### 4.7. Payment Analysis by Clusters

Table 9 shows the payment method in the corresponding clusters in different algorithms. Across each group, the customer paid on COD, Payaxis, and Easypaisa. Through these tabular data, the organization could decide to offer the payment method to their customers.

### 4.8. Product Analysis by Clusters

On the other hand, the most purchased products, ‘Mobiles and Tablets’ and ‘Men’s Fashion’, ‘Books’ and ‘School and Education’ items, were not of much interest to customers. The retailer might tailor the product recommendation based on the product research results. The summary is shown in Figure 6 for the different algorithms and their corresponding clusters.

### 4.9. Clustering Based on Financial Year

Figure 7 shows the graphical representation of the frequency, monetary, and the number of transactions for each clustering algorithm for the financial years 2017, 2018, and 2019. Most of the transactions are from FY-17 and FY-18 because FY-17 = 12 months, FY-18 = 12 months, and FY-19 = 2 months of transactions. FY-18 has the highest frequency and financial values. Only two months of FY-19 have records. Figure 7 displays the frequency and monetary values for various clusters and algorithms of the financial years 2017, 2018, 2019.

### 4.10. Clustering Based on Month-Wise and Season-Wise

Table 10 shows the month-wise frequency that occurred in the entire dataset period. The season-wise data are extracted from the dataset.

Table 11 shows the month-wise monetary that occurred in the entire dataset period. The season-wise data are extracted for monetary value.

## 5. Conclusions and Future Work

In the retail business, customer segmentation is critical. The cluster identification is an issue. Here, the question raised is which cluster is the best? For this purpose, the cluster validations were performed and the best one was elected through majority voting; i.e., 3, the stable one, was identified considering the internal cluster validation factors. Different algorithms were examined on the same feature data for segmentation using the RFMT model. Therefore, each algorithm segmented the data on its characteristics. Strong customer connections help merchants to utilize marketing resources efficiently, such as promotion strategies, pricing policies, and loyalty schemes, to maximize profits.

Initially, the records were extracted from a dataset. Then, the data took the RFMT values and translated them onto a five-centile scale as discrete scores. Finally, hierarchical, k-means, and Gaussian methods were used to divide the consumers into three groups, while DBSCAN divided the consumers into two groups.

The current segmented data will be compared, evaluated, and its accuracy verified using the suggested framework. Additionally, it can be used to determine the validity and accuracy of different datasets.

## Figures and Tables

**Figure 1 sensors-23-03180-f001:**
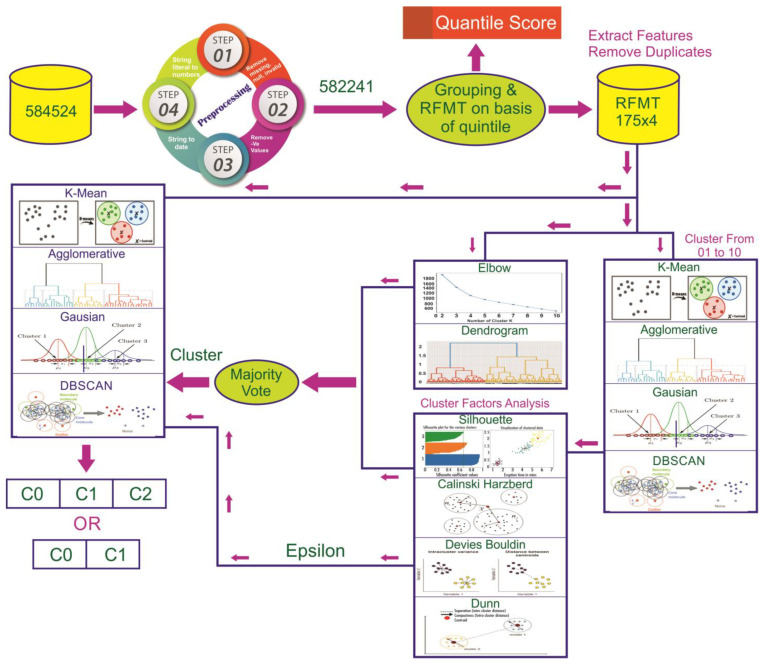
Proposed customer segmentation framework.

**Figure 2 sensors-23-03180-f002:**
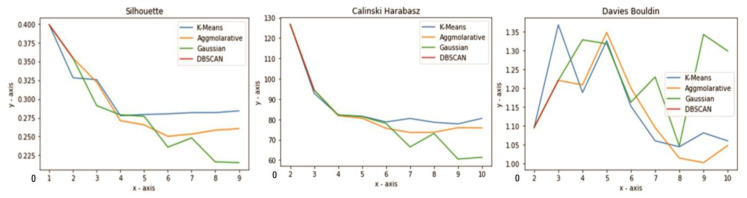
Silhouette coefficient, Calinsky–Harabasz, and Davies–Bouldin distribution on clusters for different algorithms.

**Figure 3 sensors-23-03180-f003:**
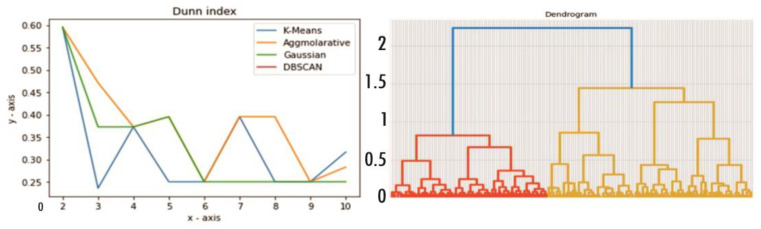
Dunn index and dendrogram for different algorithms at different cluster number.

**Figure 4 sensors-23-03180-f004:**
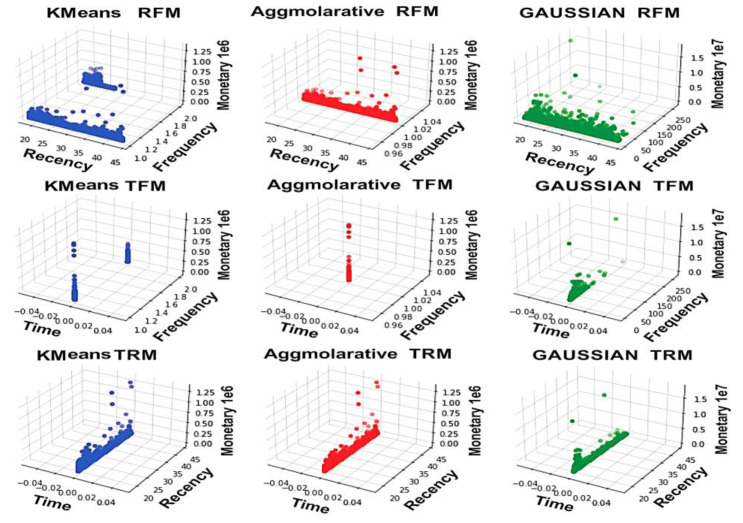
Customer distribution in the three or two clusters of RFMT in different algorithms in cluster C2.

**Figure 5 sensors-23-03180-f005:**
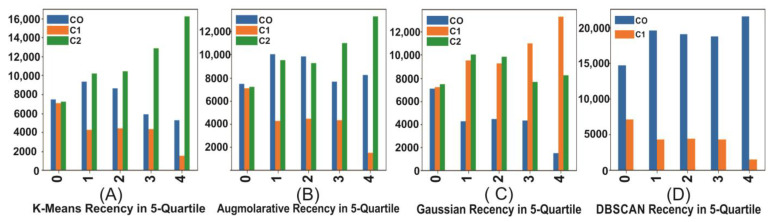
Recency distribution on a 5-quintiles/grades scale for different clusters and algorithms.

**Figure 6 sensors-23-03180-f006:**
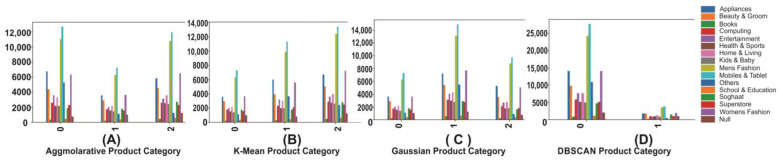
Product category distribution on 3 or 2 clusters using different algorithms.

**Figure 7 sensors-23-03180-f007:**
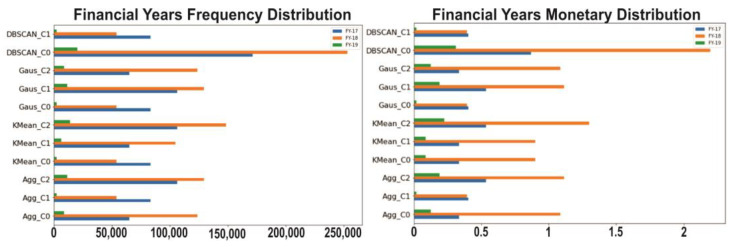
Financial year wise frequency, monetary distribution on different clusters, and algorithms.

**Table 1 sensors-23-03180-t001:** Discretized scores example for the customers.

CustomerID	Recency	Frequency	Monetary	Time	R	F	M	T
2	39.21	2	510	4	1	3	1	1
3	33.45	5	3695	10	2	4	3	1
4	18.15	428	2,748,848	25	5	5	5	1

**Table 2 sensors-23-03180-t002:** Centile upper boundary values of the RFMT variables.

RFMT Variables	20 Centile		40 Centile		60 Centile		80 Centile		>80 Centile	
	UB *	Records	UB *	Records	UB *	Records	UB *	Records	UB *	Records
Recency	23.2	21,738	27.2	23,832	31.3	23,493	37.43	23,024	4.40 × 10^1^	22,994
Frequency	1		1	50,250	2	20,826	5	24,006	2.52 × 10^3^	19,999
Monetary	999	23,813	2249	22,222	6716	23,014	26,207	23,016	3.62 × 10^7^	23,016
Time	0		0		0		0	93,445	2.50 × 10^1^	21,636

* UB: Upper Boundary Value and Records are the numbers of records.

**Table 3 sensors-23-03180-t003:** Quintile scoring values for each of the RFMT variables.

Quintile%	20	40	60	80	>80
R	5	4	3	2	1
F	4	2	3	4	5
M	3	3	3	4	5
T	2	4	4	2	1

**Table 4 sensors-23-03180-t004:** Clusters factors analysis scores of the corresponding cluster for different algorithms.

Factors	K-Means		Hierarchical		Gaussian		DBSCAN ϵ = 2.23	
	Score	Cluster	Score	Cluster	Score	Cluster	Score	Cluster
Silhouette	0.3282	3	0.3544	3	0.3544	3	0.3986	2
Dunn Index	0.2357	7	0.4714	3	0.3952	5	0.5951	2
Calinski–Harabasz	92.8496	3	94.4464	3	94.446	3	126.7604	2
Davies–Bouldin	1.0439	8	1.0017	9	1.0462	8	1.0951	2
Algorithms, wise majority voting		3		3		3		2

**Table 5 sensors-23-03180-t005:** Validation metrics of different cluster factors analysis using different algorithms for 10 clusters.

Clusters Factors	Silhouette		Calinski Harabasz		Dunn Index		Davies Bouldin		Dendrogram
Algorithms	Value	Cluster	Value	Cluster	Value	Cluster	Value	Cluster	C3
K-Means	92.85	C3	92.85	C3	0.3953	C7	1.0439	C8	
Agglomerative	94.446	C3	94.446	C3	0.4714	C3	1.0017	C9	
Gaussian	94.446	C3	94.446	C3	0.3953	C5	1.0462	C8	
DBSCAN	0.3986	C2	126.76	C2	0.5951	C2			

**Table 6 sensors-23-03180-t006:** Clusters and their frequency of occurrences.

Cluster	Frequency of Occurrences
3	7
2	4
5	1
7	1
8	2
9	1

**Table 7 sensors-23-03180-t007:** Clusters distribution, number of customers, recency, frequency, monetary, and time for different clusters and algorithms.

Models	Cluster	#Customer	Monetary	Frequency	Time	Recency
Gaussian	C0	21,636	2,292,880,342	352,683	154,370	586,924.11
	C1	50,250	367,593,227	50,250	0	1,564,783.91
	C2	43,195	1,534,777,536	198,225	0	1,277,032.86
K-means	C0	21,636	2,292,880,342	352,683	154,370	586,924.11
	C1	36,912	1,465,561,061	166,742	0	1,129,442.13
	C2	56,533	436,809,702	62,816	0	1,712,374.63
DBSCAN	C0	93,445	1,902,370,763	229,558	0	2,841,816.77
	C1	21,636	2,292,880,342	352,683	154,370	586,924.11
Agglomerative	C0	43,195	1,534,777,536	179,308	45,373	1,230,605.19
	C1	21,636	2,292,880,342	352,683	43,655	705,004.86
	C2	50,250	367,593,227	50,250	65,342	1,493,086.77

**Table 8 sensors-23-03180-t008:** Transaction status analysis value for different algorithm’s clusters.

Transaction Status	Agglomerative			DBSCAN		Gaussian			K-Means		
	C0	C1	C2	C0	C1	C0	C1	C2	C0	C1	C2
order_refunded	9366	5516	11,451	20,817	5516	5516	11,451	9366	5516	8273	12,544
complete	23,688	14,448	28,958	52,646	14,448	14,448	28,958	23,688	14,448	21,417	31,229
canceled	19,041	9074	20,817	39,858	9074	9074	20,817	19,041	9074	16,004	23,854
received	8409	2905	9066	17,475	2905	2905	9066	8409	2905	6564	10,911
closed	79	67	134	213	67	67	134	79	67	73	140
cod	274	135	292	566	135	135	292	274	135	223	343
fraud	2	2	6	8	2	2	6	2	2	2	6
\N or Null	0	0	1	1	0	0	1	0	0	0	1

**Table 9 sensors-23-03180-t009:** Payment analysis for different algorithm clusters.

Payment	Agglomerative			DBSCAN		Gaussian			K-Means		
	C0	C1	C2	C0	C1	C0	C1	C2	C0	C1	C2
COD	26,399	15,857	33,648	60,047	15,857	14,655	33,648	26,399	15,857	23,350	36,697
customercredit	877	654	1179	2056	654	560	1179	877	654	797	1259
Easypay	8381	2843	8365	16,746	2843	2751	8365	8381	2843	6797	9949
Payaxis	7163	4929	8672	15,835	4929	4723	8672	7163	4929	6560	9275

**Table 10 sensors-23-03180-t010:** Month-wise frequency value for different algorithm’s clusters.

Months		Agglomerative			K-Means			Gaussian			DBSCAN	
		C0	C1	C2	C0	C1	C2	C0	C1	C2	C0	C1
1	F	7569	6074	12,420	6074	6632	13,357	6074	12,420	7569	19,989	6074
2	F	14,495	7338	16,944	7338	11,499	19,940	7338	16,944	14,495	31,439	7338
3	F	23,576	9313	28,593	9313	18,093	34,076	9313	28,593	23,576	52,169	9313
4	F	9860	8234	15,997	8234	8431	17,426	8234	15,997	9860	25,857	8234
5	F	22,579	14,166	25,858	14,166	19,934	28,503	14,166	25,858	22,579	48,437	14,166
6	F	11,754	7380	15,396	7380	10,310	16,840	7380	15,396	11,754	27,150	7380
7	F	11,820	9972	17,359	9972	10,559	18,620	9972	17,359	11,820	29,179	9972
8	F	14,650	12,611	20,759	12,611	13,470	21,939	12,611	20,759	14,650	35,409	12,611
9	F	5650	8812	9562	8812	5650	9562	8812	9562	5650	15,212	8812
10	F	8410	9690	12,523	9690	8399	12,534	9690	12,523	8410	20,933	9690
11	F	57,702	38,549	59,205	38,549	54,975	61,932	38,549	59,205	57,702	116,907	38,549
12	F	9221	7265	12,713	7265	7967	13,967	7265	12,713	9221	21,934	7265

**Table 11 sensors-23-03180-t011:** Month-wise monetary value for different algorithm clusters.

Months		Agglomerative			K-Means			Gaussian			DBSCAN	
		C0	C1	C2	C0	C1	C2	C0	C1	C2	C0	C1
1	M	52,817,070	35,497,252	83,007,712	35,497,252	46,071,829	89,752,953	35,497,252	83,007,712	52,817,070	135,824,782	35,497,252
2	M	142,698,482	70,539,371	158,086,835	70,539,371	119,341,432	181,443,885	70,539,371	158,086,835	142,698,482	300,785,317	70,539,371
3	M	194,171,895	67,220,732	234,976,024	67,220,732	150,578,305	278,569,614	67,220,732	234,976,024	194,171,895	429,147,919	67,220,732
4	M	52,060,438	48,470,683	104,216,039	48,470,683	42,756,751	113,519,726	48,470,683	104,216,039	52,060,438	156,276,477	48,470,683
5	M	239,407,852	106,884,218	268,887,881	106,884,218	195,297,204	312,998,529	106,884,218	268,887,881	239,407,852	508,295,733	1.07 × 10^8^
6	M	142,683,774	49,594,035	134,496,783	49,594,035	114,788,239	162,392,318	49,594,035	134,496,783	142,683,774	277,180,557	49,594,035
7	M	103,068,273	34,418,189	161,960,476	34,418,189	79,184,939	185,843,810	34,418,189	161,960,476	103,068,273	265,028,749	34,418,189
8	M	86,746,112	43,933,626	122,875,593	43,933,626	74,822,254	134,799,451	43,933,626	122,875,593	86,746,112	209,621,705	43,933,626
9	M	31,958,211	47,621,531	54,922,810	47,621,531	31,958,211	54,922,810	47,621,531	54,922,810	31,958,211	86,881,021	47,621,531
10	M	49,059,185	60,600,756	74,950,374	60,600,756	49,051,180	74,958,379	60,600,756	74,950,374	49,059,185	124,009,559	60,600,756
11	M	406,926,351	214,439,839	371,593,983	214,439,839	381,310,046	397,210,288	214,439,839	371,593,983	406,926,351	778,520,334	2.14 × 10^8^
12	M	41,771,599	33,043,018	63,723,613	33,043,018	34,742,549	70,752,663	33,043,018	63,723,613	41,771,599	105,495,212	33,043,018

## Data Availability

The data presented in this study are available on request from the corresponding author.

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
