# Peer review of "Customer Analysis Using Machine Learning-Based Classification Algorithms for Effective Segmentation Using Recency, Frequency, Monetary, and Time"

_sensors, 2023, doi:10.3390/s23063180_

Round 1

Author Response

Dear Reviewer,

Thank you so much for the quality comments which strengthen our manuscript. Your all concerns have been addressed and point by point reply along with suggested changes have been mentioned in the Response to Reviewers file. All the changes have also been addressed in the revised manuscript. 

Regards,

Authors

Reviewer 2 Report

This paper used different clustering approaches for analyzing Pakistan’s e-commerce dataset.

The are several limitations :

·        What are the contribution and originality?

·        There is no literature review on ML approaches in customer segmentation analysis.

·        What are the gaps?

·        How does the paper fill the gaps?

Author Response

(The authors gave the same response as above.)

Reviewer 3 Report

*Why author choose RFMT?

* Abstract need to rewite..

* Pakistan e-commerce largest dataset is not keyword.

* why Upper Boundary Value needed?

* Use of Dendrogram diagram and What author infer from this diagarm?

* Author choose only four Machine Learning Models ? Recent for choosing only particular model to this applicatin?

*  How Table 8 & Table 9 used for author result?

Author Response

(The authors gave the same response as above.)

Reviewer 4 Report

The authors study Customer Analysis Using Machine Learning-Based Classification Algorithms for Effective Segmentation. The following cases are obligatory to review the paper more robustly.

a)     The technical term “RFMT” is not recommended in the title.

b)     The COVID-19 seems to have disappeared globally, and it is recommended to remove the relevant research background.

c)      The author's contribution should be based on the literature review and further analysis.

d)     The citation of some references is incorrect, for example, [25], [26], [27], [3] in Section 2.4. “Author [number]” is recommended.

e)     The word “K-Means” in Figure 1 is inconsistent with other parts of the manuscript. Please carefully correct such problems.

f)      As for the determination of clustering number, please refer to the published paper [10.1016/j.jclepro.2022.132734].

For the main reasons stated above, this paper should be revised and resubmitted.

Author Response

(The authors gave the same response as above.)

Round 2

Reviewer 2 Report

The problem with this study is the lack of deep insight into the literature on customer segmentation.

As mentioned before, originality cannot be limited to using different clustering algorithms; it should be based on the results' interpretation and the customers' strategic profiling. 

What are the gaps in the existing literature review?

What are the implications of this article for filling the gaps?

Author Response

Cover Letter – (Responses to Reviewers’ Comments)

Paper title: Customer Analysis Using Machine Learning-Based Classification Algorithms for Effective Segmentation Using Recency, Frequency, Monetary and Time

Manuscript ID: 2232557

To: Editor

Re: Response to Reviewer’s

Dear Editor,

Thank you again for allowing a resubmission of our manuscript, with an opportunity to address the reviewers’ valuable comments. “Customer Analysis Using Machine Learning Based Classification Algorithms for Effective Segmentation using RFMT – 2232557”. We duly checked and addressed the reviewers’ concern(s)/question(s) and comments, explained below in an itemized fashion.

We are uploading (a) our point-by-point response to the comments (below) (response to reviewers), (b) an updated manuscript.

Best regards,

Dr. Inayat Khan

Submitting Author

Concern   #   1: What are the gaps in the existing literature review

Response # 1: We are thankful to the reviewer for their efforts to review the paper and offer valuable suggestions. We are agreed with the reviewer’s concern.

ACTION: The research gaps in the literature have been properly investigated, and their discussion has been added and highlighted in the revised manuscript. The RFM (Recency, Frequency, and Monetary) analysis is a widely used approach in customer segmentation. But the issue is that it does not consider the time factor. Thus, our research is viable to investigate the inclusion of time (T) in the RFMT model to understand customer loyalty and customer behavior better. Similarly, there is no validation or validation on only one criterion, which may be biased or produce biased results. The literature review suggests that different cluster validation factors have been used to validate the clusters (Silhouette, Calinski Harabasz, Dunn index, Davies Bouldin and Dendrogram), but there is no agreement that which factor is the most effective. Therefore, further research could compare and evaluate the performance of different cluster validation factors to identify the most accurate and stable cluster. The literature review highlights the challenge of choosing the best segmentation approach due to the different characteristics of the algorithms. Thus, our research could investigate the use of majority voting, an ensemble method that combines multiple clustering algorithms, to improve the accuracy and stability of customer segmentation.

Concern   #   2: what are the implications of this article for filling the gaps?

Response # 2: We thank the reviewer for their efforts and time in reviewing the paper and offer valuable suggestions. We are agreed with the reviewer’s concern.

ACTION: To fulfil the research gaps as mentioned in the literature, in this research article, we used cluster validation criteria to verify the cluster’s validity, majority voting to select the cluster, and different algorithms for segmentation using RFMT model. Our research is viable to investigate the inclusion of time (T) in the RFMT model to understand customer loyalty and customer behavior. To overcome the validation issue and fill this gap for producing meaningful knowledge/ results, we have used multiple cluster validation factors to enhance the cluster stability, as shown in Table 5 in the revised manuscript. To fill the research gap in choosing the best segmentation approach, we have investigated the use of majority voting, an ensemble method that combines multiple clustering algorithms, to improve the accuracy and stability of customer segmentation.

Reviewer 4 Report

The author spent a lot of time and energy to improve the manuscript, and the quality of the article has been significantly improved. I think it can be published

Author Response

Dear anonymous Reviewer,

Thank you again for your time and efforts to review the article.

Regards,